# Comparing preferences to evaluations of barrier self-efficacy for two strength training programs in US older adults

Jordan D. Kurth[1,2]*, Christopher N. Sciamanna[1,2], Cheyenne Herrell[1,2], Matthew Moeller[1,2], Jonathan G. Stine[3]

1 Penn State University, University Park, PA, United States of America, 2 Penn State College of Medicine, Hershey, PA, United States of America, 3 Penn State Health–Milton S. Hershey Medical Center, Hershey, PA, United States of America

* jdk5930@psu.edu

## Abstract

### Background/Objectives

Engagement in regular physical activity is one of the best strategies for older adults to remain healthy. Unfortunately, only 35% of older adults meet guidelines for muscle strengthening activities. Eliciting participant preferences is one possible way to improve physical activity engagement. However, other sources of participant input to improve uptake and maintenance remain uninvestigated. This study compared preferences to self-efficacy ratings for two strength training programs.

### Methods

We conducted a national cross-sectional survey of 611 US adults over age 65. We compared two participant evaluations (the preferred program and the program for which they had higher barrier self-efficacy) of two hypothetical strength training programs (45 minutes performed three times per week (traditional) and 5 minutes performed daily (brief)).

### Results

Most participants (68%) preferred the brief strength training program. The difference in self-efficacy ratings was an average of 1.2 (SD = 0.92). One in five participants preferred a strength training program for which they had less self-efficacy; nearly all of these participants (92%) preferred the traditional strength training program but had more self-efficacy for the brief strength training program.

### Conclusion

Older adults reported preferring and having more self-efficacy for a brief compared to a traditional strength training program. Differences in self-efficacy ratings between the two strength training programs were large. Preferences were often not congruent with ratings of self-efficacy.

**Data Availability Statement:** Data associated with this study is available via Penn State Data Commons at https://doi.org/10.26208/Q1KB-AW67.

**Funding:** The author(s) received no specific funding for this work.

**Competing interests:** I have read the journal's policy and the authors of this manuscript have the following competing interests: Christopher Sciamanna has an investment, such as stock, in a company which has begun to investigate the possibility of creating a business that provides exercise programs. All other authors declare that they have no known competing financial interests or personal relationships that could have appeared to influence the work reported in this paper.

## Significance/Implications

Preferences for strength training programming may not always reflect the program most likely to be maintained. Future investigations should evaluate differences in behavioral uptake, maintenance, and outcomes from two comparative strength training interventions using preferences and self-efficacy.

## Introduction

Engagement in regular physical activity (PA) is one of the best strategies for adults over the age of 65 to preserve mobility, reduce falls risk, and remain both physically and mentally healthy [1–5]. Unfortunately, 31% of U.S. older adults report performing zero physical activity in their leisure time, and only 23% report meeting both the Centers for Disease Control and Prevention guidelines for aerobic activity (150 minutes of moderate-to-vigorous physical activity per week) and the recommendations for muscle strengthening activities (two days per week) [6]. Thus, a need exists to enhance current physical activity promotion efforts among older adults.

One possible physical activity promotion effort is to align characteristics (e.g., duration, frequency, mode of physical activity, reward structure) of physical activity programs with the preferences of the participants. There have been numerous examples of physical activity preference identification, particularly in community-dwelling older adults and clinical populations, such as those with chronic joint pain [7–10]. Despite the intuitive appeal of this approach, there is limited evidence to support its influence on increasing regular physical activity engagement [11–13]. Additionally, preferences are often operationally defined as "a predisposition to like a particular type of or context for physical activity more than others *and to choose it when given the opportunity* [emphasis added]" [14]. This definition, and much of the evidence that does exist surrounding the influence of preference alignment in physical activities, often evaluates self-selection [12,13] instead of preference alignment [11]. While the connection between allowing an individual to self-select program characteristics and improved affective response is well-established [15,16], this is of limited value when attempting to design a large, prescribed, scalable physical activity program. What we do know about expressed preferences generally suggests that they evolve across social and environmental contexts, and in some cases may be difficult to articulate prior to a relevant experience [17]. Additionally, they may be susceptible to repeated exposure to or familiarity with the stimuli being presented [18–20], thereby calling into question their utility in informing program design, particularly with relatively low-active populations that are frequently targeted.

The goal of most physical activity programming is long-term engagement in the behavior, or maintenance, as repeated performance is required to reap the benefits associated with physical activity. Identifying factors associated with maintenance of regular physical activity–and in fact even defining the phenomenon of "maintenance" itself–remains an active pursuit of both research and clinical practice [21]. However, of what is known, self-efficacy (SE; the level of confidence in one's capability to engage in the behavior) is to-date perhaps the single most well-established predictor of physical activity engagement and maintenance according to systematic reviews and meta-analyses [22–25]. As such, designing programs that maximize known predictors of long-term physical activity engagement such as self-efficacy may promote long-term behavior performance. One novel method by which this might be accomplished is to use the evaluation of program characteristics rated using a known predictor of physical activity maintenance (i.e., self-efficacy) in place of the traditional rating of preference [26].

Therefore, this study aimed to compare two hypothetical evaluations: [1] preferred strength training program (5-minute daily program vs. 45-minute program performed 3 times per week) and [2] barrier self-efficacy for each respective program in a population of adults over the age of 65 years. We hypothesized that most participants would prefer the shorter, more frequent training program, and have higher self-efficacy for that same program. We also hypothesized that most participants would report preferring a program for which they had higher self-efficacy.

## Materials and methods

This institutional review board exempted study (Penn State University IRB# STUDY00022306) used an anonymous online cross-sectional survey to inform the design of strength training programs in adults over 65 years of age with varying levels of health status and comorbidities. Throughout this manuscript, the reporting of data are in-line with the Strengthening the Reporting of Observational Studies in Epidemiology (STROBE) guidelines for observational cross-sectional studies [27]. The survey was conducted using a commercial survey company (Qualtrics). Qualtrics uses a network of survey participants from many suppliers with a range of recruitment methodologies from across the US. Participants are sourced from different methods depending on the supplier, including advertisements and promotions on smartphones, referrals from membership lists, social networks, mobile games, banner advertisements, mail-based recruitment campaigns and others [28,29]. The survey was estimated to take 5–10 minutes to complete and all participants were compensated. To ensure data quality, surveys included [1] attention checks (i.e., factual questions with correct answers) and [2] speeding checks (i.e., eliminating responses from those who completed the online survey in less than one-third the median duration of survey completion).

### Participants

Participants were required to be at least 65 years of age, located in the United States, and fluent in the English language. Additional target demographic quotas were placed on recruitment to ensure a sample representative of the population over 65 years of age in the US. Target demographic quotas were placed on sex (50% male/50% female) and race/ethnicity (55% non-White Hispanic, 15% African American, 15% Asian American, 15% Alaskan Native/Native Hawaiian; 10% Hispanic). A target minimum of 556 participants (278 per preferred program) were planned to be enrolled in order to be able to detect a small (Cohen's $d$ = 0.2) difference at 90% power at an $\alpha$ = 0.05 within each preference group. Data collection occurred between April 3, 2023 and May 31, 2023. The need for consent was waived by the institutional review board, as data was collected anonymously.

### Instruments and measures

**Demographic information.**   Smoking status was assessed using a single item adapted from the Behavioral Risk Factor Surveillance System (BRFSS), *"Do you use a tobacco product every day?"* [30]. Medical history was assessed using questions (also from BRFSS) assessing presence or absence of: diabetes, high cholesterol, heart disease, osteoporosis, hypertension, stroke, and arthritis, as well as the frequency of strength training [30]. Demographic and anthropometric characteristics, such as age, gender, race, ethnicity, height, and weight were assessed, and body mass index (BMI) was calculated using the standard formula.

**Strength training program preference.**   Strength training program preference was assessed by asking the following, "*We are designing a strength training program for people over 65 to use at home, to improve their physical function, ability to walk and to reduce falls. Assuming that all programs include similar types of exercises, which would you prefer?*" Participants

were given two strength training program options from which to choose: *"5 minutes, every day"* or *"45 minutes, 3 times per week"*. The order of these answer options was randomized for each participant.

**Strength training program self-efficacy.** Self-efficacy for each strength training program was assessed using a 5-item, 5-point Likert scale (*Not at all confident* to *Extremely confident*) [31] to respond to the prompt *"Regardless of your previous selection, please answer the following regarding a strength training program performed [45 minutes, 3 days per week OR 5 minutes, every day]. How confident are you that you could complete this program under each of the following conditions over the next 12 months? I could exercise for [45 minutes, 3 times per week OR 5 minutes, each day. . .]"*. The order of the presentation of each program's scale was also randomized for each participant. The five items evaluated by each participant were: *"when I am tired.", "when I am in a bad mood.", "when I feel I do not have the time.", "when I am on vacation.",* and *"during bad weather (i.e., raining or snowing).".*

## Analysis

Participants were grouped by their expressed strength training program preference. Additionally, two mean self-efficacy scores, one for the 5-minute daily program and one for the 45-minute program performed three days per week, were then calculated for each participant. These two scores were then compared; where applicable, the program for which participants reported higher SE was identified as the program for which the participant had the most self-efficacy. The distribution of these two variables (program preference and higher/lower/equal self-efficacy) were then cross-tabulated for comparison.

## Results

### Participant profile

A total of 1,162 of the participants that were screened met inclusion criteria. Data from participants that did not complete the survey (n = 438) were excluded from the final sample. Additionally, participants whose demographic quotas were filled before they completed their survey (n = 95), that completed the survey too quickly (less than half of the median complete time; n = 9), that failed the attention check (n = 5), or that were identified as bots using embedded survey fields (n = 4) were removed. Therefore, the final sample size included 611 participants (Fig 1). A demographic summary of the 611 participants that completed the survey is provided in Table 1. Of note, the mean participant age was 72 years. Approximately half of participants were female. Racial and ethnic distributions were representative of the US population over the age of 65 [32].

### Description of strength training program preference and self-efficacy

Most participants (68%) expressed preference for the 5-minute daily strength training program over the 45-minute program completed three times per week. Even more participants (80%) reported having more self-efficacy for the 5-minute, daily program compared to the 45-minute program completed three days per week. The mean absolute difference in program self-efficacy rating was 1.2 (SD = 0.92; Cohen's *d* = 1.3).

### Congruence of strength training program preference and self-efficacy rating

Most participants' (402/611; 66%) expressed strength training program preference was congruent with the program for which they had the most self-efficacy. However, more than 1 in 5

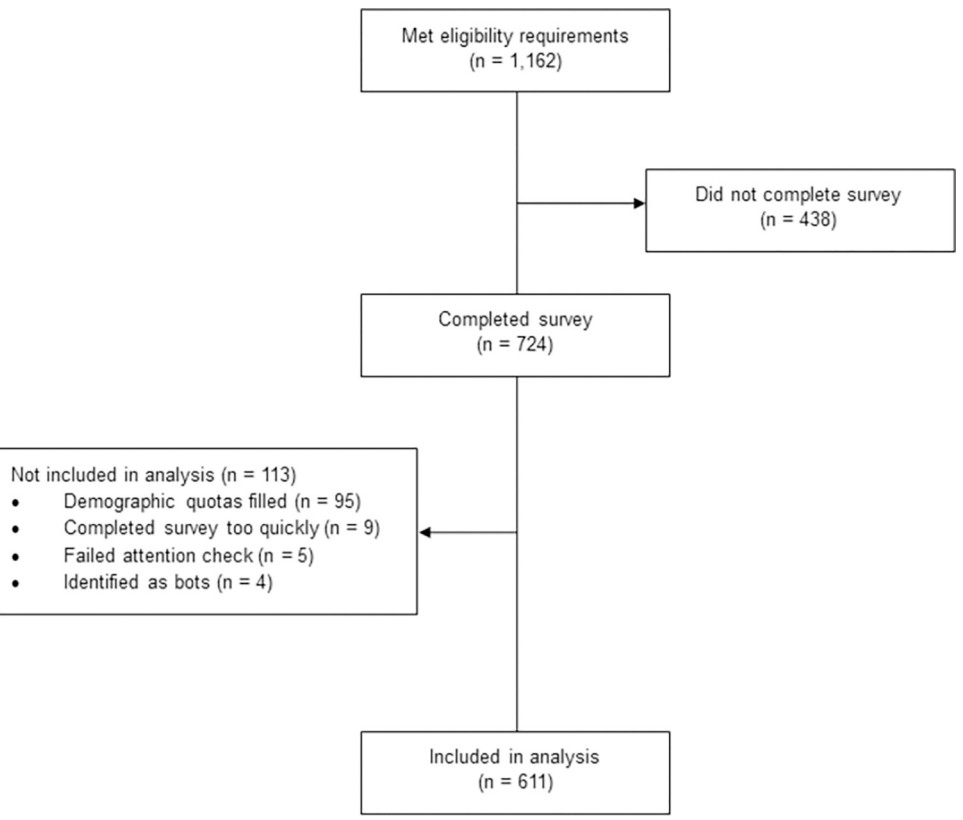

**Fig 1. Participant flow diagram.**

**Table 1. Participant characteristics.**

|  | Total (N = 611)<br>Mean(SD) or n(%) |
| --- | --- |
| Age, years | 72.4 (5.1) |
| Sex, female, n | 294 (48.1) |
| Race, White, n | 341 (55.8) |
| Race, Black, n | 102 (16.7) |
| Ethnicity, Hispanic, n | 62 (10.2) |
| Body Mass Index, kg/m$^2$ | 27.7 (7.0) |
| Strength Training, days/week | 1.4 (2.1) |
| Daily Tobacco Use, n | 64 (10.5) |
| Diabetes, n | 152 (25.2) |
| High Cholesterol, n | 338 (56.5) |
| Heart Disease, n | 787 (14.6) |
| Osteoporosis, n | 100 (17.0) |
| Hypertension, n | 375 (62.1) |
| Stroke, n | 40 (6.6) |
| Arthritis, n | 295 (49.4) |
| Prefer 5-min, daily program, n | 380 (68.4) |
| Higher SE for 5-min, daily program, n | 490 (80.2) |

*Note.* SE = self-efficacy.

**Table 2. Cross-tabulation of preference and higher self-efficacy rating.**

| | Self-Efficacy Rating Higher for 45-minute, 3 Times per Week | Self-Efficacy Rating Higher for 5-minute, Daily | Self-Efficacy Rating Equal for Both Programs |
|---|---|---|---|
| Prefer 45-minute, 3 Times per Week Program | 32 (5.2%) | 120 (19.6%) | 41 (6.7%) |
| Prefer 5-minute, Daily Program | 10 (1.6%) | 370 (60.6%) | 38 (6.2%) |

*Note*. Percentages are relative to the total sample (N = 611).

participants (130/611; 21.3%) expressed a strength training program preference that was incongruent with the program for which they reported higher self-efficacy. Nearly all (370/418; 89%) participants who reported preferring the 5-minute daily program also had higher self-efficacy for that same program, whereas only 17% of participants (32/193) who reported preferring the 45-minute program completed 3 days per week also had higher self-efficacy for the 45-minute program. Approximately 13% (79/611) participants reported the same self-efficacy for both programs, reporting nearly equal preferences for each (n = 41 vs. n = 38; Table 2).

## Discussion

In this study, consistent with our hypothesis, more than 2 in 3 adults over the age of 65 expressed a preference for a 5-minute daily strength training program compared to a 45-minute program completed three times per week. Also consistent with our hypothesis, more than 80% of participants reported higher self-efficacy for the 5-minute program. Two in three participants preferred the program for which they had higher self-efficacy. However, more than four out of every five participants that preferred the 45-minute program had higher self-efficacy for the 5-minute program.

The present study suggests that elicited preferences are often not congruent with ratings of self-efficacy. The impact of this discrepancy between preference selection and higher self-efficacy ratings on long-term behavior remains unknown. Given that most participants who preferred the 45-minute program had higher self-efficacy for the 5-minute program, and that the 45-minute program structure is the most commonly offered, the potential impact of designing self-efficacy driven programs could be large. What is known is that the literature supporting the positive effects of self-efficacy [22–25] is much stronger than the literature supporting the effects of preference selection [11–13].

Additionally, preferences are known to not necessarily arise directly from cognitions [20]. In other words, they are rife with influence from irrelevant and/or counterproductive sources–such as emotional states and repeated exposure to or familiarity with the stimuli being presented [18–20]. Standard physical activity programs are typically 45–60 minutes performed three days per week. This is one possible factor that may have led some participants to report preferring the 45-minute program despite having higher self-efficacy for the 5-minute program. While self-efficacy is not immune to emotional influence [33], that emotional influence is theoretically more directly relevant to the activity in question. For those reasons, soliciting evaluations of possible program options on known predictors of the target behavior (in this case self-efficacy for physical activity maintenance) instead of preference may lead to more effective decision-making by intervention designers targeting physical activity maintenance.

This is a preliminary investigation into asking for program evaluations using a known predictor of the target behavior (in this case, barrier self-efficacy for physical activity) instead of preference. This study is not intended to suggest that a 5-minute daily strength training

workout is the best strength training format for older adults or that it is better than a 45-minute program completed three times per week. Further exploration into alternative/additional program details, the number of options presented, and how those programs are described is required. Most importantly, investigation also needs to be done to evaluate differences in actual behavioral uptake, maintenance, and outcomes that arise from two comparative interventions: one designed using participant preferences and one designed using a known predictor of maintenance.

This study has several limitations. First, only two program options were evaluated; information about any alternative program options that may be preferable or elicit higher self-efficacy evaluations is not available. Second, the relevance of the difference in self-efficacy, though large (Cohen's $d$ = 1.3), is unclear. While one program was often rated higher than the other and dichotomized as the program for which self-efficacy was higher, the magnitude of that difference varied by participant. Finally, the impact of preference and self-efficacy evaluations was not evaluated by assessing actual behavior uptake, maintenance, or outcomes. It remains unclear if there is a behavioral or outcome-based difference between preference and self-efficacy-designed programs.

This study also has multiple strengths. Sampling quotas allowed for a demographically-representative sample of the US population over the age of 65 years based on the characteristics of sex, race, and ethnicity–overcoming a well-documented limitation of exercise promotion research historically [34]. Additionally, data was collected using a digital, anonymous survey, thereby reducing–though not eliminating–self-selection bias common in physical activity research. Additionally the presentation of the self-efficacy questions and preference answer options were presented randomly to eliminate order effects.

## Conclusion

In this novel study directly comparing preferences and self-efficacy, adults over the age of 65 years reported preferring a 5-minute daily strength training program compared to a 45-minute program performed three times per week. Most participants also reported higher self-efficacy for that program. However, 1 in 5 older adults preferred a program for which they had lower self-efficacy than the alternative; nearly all of these participants preferred the 45-minute program but had lower self-efficacy for it. This evidence suggests that expressed preferences for physical activity programming may not always reflect the program most likely to be maintained. Soliciting evaluations of possible program options on known predictors of the target behavior (in this case self-efficacy for physical activity maintenance) instead of preference may lead to more effective decision-making by intervention designers targeting physical activity maintenance.

## Supporting information

**S1 Checklist. STROBE statement—checklist of items that should be included in reports of *cross-sectional studies*.**
(DOCX)

## Author Contributions

**Conceptualization:** Jordan D. Kurth, Christopher N. Sciamanna, Cheyenne Herrell, Matthew Moeller, Jonathan G. Stine.

**Formal analysis:** Jordan D. Kurth, Christopher N. Sciamanna.

**Methodology:** Jordan D. Kurth, Christopher N. Sciamanna.

**Project administration:** Jordan D. Kurth, Christopher N. Sciamanna.

**Visualization:** Jordan D. Kurth.

**Writing – original draft:** Jordan D. Kurth, Christopher N. Sciamanna, Cheyenne Herrell, Matthew Moeller, Jonathan G. Stine.

**Writing – review & editing:** Jordan D. Kurth, Christopher N. Sciamanna, Cheyenne Herrell, Matthew Moeller, Jonathan G. Stine.

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
