## [Decision Letter · Decision Letter 0]

13 Mar 2024

PONE-D-24-04115Comparing Preferences to Evaluations of Barrier Self-Efficacy for Two Strength Training Programs in US Older AdultsPLOS ONE

Dear Dr. Kurth,

Thank you for submitting your manuscript to PLOS ONE. After careful consideration, we feel that it has merit but does not fully meet PLOS ONE’s publication criteria as it currently stands. Therefore, we invite you to submit a revised version of the manuscript that addresses the points raised during the review process.

We look forward to receiving your revised manuscript.

Kind regards,

Sohel Ahmed, BPT, MPT, MDMR

Academic Editor

PLOS ONE

Additional Editor Comments:

• Why is this study really important? The Centers for Disease Control and Prevention say that adults aged 65 and up need to do at least 150 minutes of moderate-intensity exercise or 75 minutes of vigorous-intensity activity each week. This could be broken down into 30 minutes of activity each day, five days a week. So why do you say that people should only work out for 35 minutes a week?

• What is the implication of this study for clinical practices?

• How do you calculate the appropriate sample size for this study? What was the sampling method you used? How did the study become a national representative survey?

• As this study is an online survey, how do you calculate the potential response rate?

• A third party conducted this survey; how can you reduce unresponsive bias?

• A lot of people might not be able to use the internet or fill out an online survey because they are not tech-savvy enough. How will they be a part of the study? How can you say that the survey is representative of the whole country?

• Inclusion and exclusion criteria should be more specific

• This survey is focusing on the efficiency of 5 minutes and 45 minutes of strength training for US older adults; why not 10/15/20 and 45 minutes?

• Discuss the generalizability of the study findings

• This study should acknowledge the potential bias of the study.

Reviewers' comments:

Reviewer's Responses to Questions

**Comments to the Author**

1. Is the manuscript technically sound, and do the data support the conclusions?

Reviewer #1: Yes

Reviewer #2: Partly

2. Has the statistical analysis been performed appropriately and rigorously? 

Reviewer #1: No

Reviewer #2: Yes

3. Have the authors made all data underlying the findings in their manuscript fully available?

Reviewer #1: Yes

Reviewer #2: No

4. Is the manuscript presented in an intelligible fashion and written in standard English?

Reviewer #1: Yes

Reviewer #2: Yes

5. Review Comments to the Author

Reviewer #1: Thank you for inviting me to review this paper. This paper is well written and very interesting and important for the readers. However, there are some minor changes required to improve quality of this paper.

Introduction

The introduction section should be elaborated by reflecting the significance and novelty of the study.

Methodology

What were the exclusion criteria? Who did the survey?

The statistical analysis section should reflect all the tests used.

Discussion

In discussion section, focus more on potential implications of your study's findings and add future research direction based on current findings.

Conclusion

Clearly state the contributions of your study to the field and highlight any novel aspects of your research. Future work should be mentioned in discussion section.

Language and Writing Style

The manuscript requires careful proofreading to eliminate grammatical errors and improve sentence structure. Clarity and precision in language will significantly enhance the manuscript's readability and overall impact.

I appreciate your attention to these matters and look forward to reviewing the revised version of your manuscript.

Reviewer #2: The study addresses an important public health issue of physical activity among older adults. As a whole, it is a well-structured manuscript. However, the main issue of the manuscript is the lack of detail in the methods section, which needs to be corrected.

A point of ambiguity throughout the text is the procedure for choosing the type of strength training, the kinds of contractions, the duration of contractions, and its specifics. In the case of older adults, the choice of resistance exercise has unique details that are not mentioned anywhere in the study. Was the exercise instruction also provided online? How can the authors ensure that the participants correctly learned and executed the exercise? The study does not provide information on how the strength training programs were presented to the participants, which could influence their preferences and self-efficacy ratings.

The study does not provide details on how the survey questions and the 5-point Likert scale were developed or validated. This information is crucial to assess the reliability and validity of the findings.

In general, the methods section lacks clarity, and many details are missing. The methods should be written in a way that allows the audience to replicate the study, but the necessary details are not presented.

6. PLOS authors have the option to publish the peer review history of their article (what does this mean?). If published, this will include your full peer review and any attached files.

Reviewer #1: No

Reviewer #2: **Yes: **Sahar Boozari

---

## [Author Response · Author response to Decision Letter 0]

19 Mar 2024

Editor Comments

Why is this study really important? The Centers for Disease Control and Prevention say that adults aged 65 and up need to do at least 150 minutes of moderate-intensity exercise or 75 minutes of vigorous-intensity activity each week. This could be broken down into 30 minutes of activity each day, five days a week. So why do you say that people should only work out for 35 minutes a week?

Thank you for raising this point. This is the recommended amount of exercise; however, less than 3 in 10 older adults are able to reach this amount. More than 3 in 10 do absolutely zero exercise. We do not suggest in this manuscript that people should only exercise for 35 minutes per week – however, increasing exercise from 0 minutes per week to any non-zero amount per week is in line with the lay recommendation to “move more, sit less”. This point is illustrated in the quotes below:

Introduction:

“Unfortunately, 31% of U.S. older adults report performing zero physical activity in their leisure time, and only 23% report meeting both the Centers for Disease Control and Prevention guidelines for aerobic activity (150 minutes of moderate-to-vigorous physical activity per week) and the recommendations for muscle strengthening activities (two days per week)(6). Thus, a need exists to enhance current physical activity promotion efforts among older adults.”

Discussion:

“This study is not intended to suggest that a 5-minute daily strength training workout is the best strength training format for older adults or that it is better than a 45-minute program completed three times per week.”

What is the implication of this study for clinical practices?

Thank you for raising this question. This study is meant to inform the design of exercise interventions. To the extent that clinical practice is involved in exercise program design, this study implies that patients may not always prefer the program they are most confident they can stick with. This point is illustrated in the quote below:

Discussion:

“For those reasons, soliciting evaluations of possible program options on known predictors of the target behavior (in this case self-efficacy for physical activity maintenance) instead of preference may lead to more effective decision-making by intervention designers targeting physical activity maintenance.”

How do you calculate the appropriate sample size for this study? What was the sampling method you used? How did the study become a national representative survey?

The following information has been added regarding sample size calculation in the Materials and Methods section:

“A target minimum of 556 participants (278 per preferred program) were planned to be enrolled in order to be able to detect a small (Cohen’s d = 0.2) difference at 90% power at an α = 0.05 within each preference group.”

Qualtrics Survey Software conducted the distribution of the survey, as indicated in this quote from the Materials and Methods section:

“The survey was conducted using a commercial survey company (Qualtrics). Qualtrics uses a network of survey participants from many suppliers with a range of recruitment methodologies from across the US. Participants are sourced from different methods depending on the supplier, including advertisements and promotions on smartphones, referrals from membership lists, social networks, mobile games, banner advertisements, mail-based recruitment campaigns and others (28,29).”

Quota sampling was employed in order to ensure a nationally-representative sample based on the criteria of sex, race, and ethnicity of people over 65 years of age in the United States. This is illustrated by the quote from the Materials and Methods section below:

“Additional target demographic quotas were placed on recruitment to ensure a sample representative of the population over 65 years of age in the US. Target demographic quotas were placed on sex (50% male/50% female) and race/ethnicity (55% non-White Hispanic, 15% African American, 15% Asian American, 15% Alaskan Native/Native Hawaiian; 10% Hispanic).”

As this study is an online survey, how do you calculate the potential response rate?

Thank you for asking for this clarification. It is not possible to quantify the number of people who had the potential to respond to the survey (i.e., those that received invitation from Qualtrics, but did not click the link to start the survey). However, Qualtrics software collects information about all responses that were started, even those not finished, so we do have information about the completion rate of the survey, as below from the Results section:

“A total of 1,162 of the participants that were screened met inclusion criteria. Data from participants that did not complete the survey (n = 438) were excluded from the final sample. Additionally, participants whose demographic quotas were filled before they completed their survey (n = 95), that completed the survey too quickly (less than half of the median complete time; n = 9), that failed the attention check (n = 5), or that were identified as bots using embedded survey fields (n = 4) were removed. Therefore, the final sample size included 611 participants (Figure 1).”

A third party conducted this survey; how can you reduce unresponsive bias?

Thank you for highlighting this. The third party uses business with whom participants already have an existing relationship to deliver the survey. The survey was estimated to take 5-10 minutes. All participants were compensated. All of these factors mitigate nonresponse bias. This information has been added to the Materials and Methods section, as quoted below:

“The survey was conducted using a commercial survey company (Qualtrics). Qualtrics uses a network of survey participants from many suppliers with a range of recruitment methodologies from across the US. Participants are sourced from different methods depending on the supplier, including advertisements and promotions on smartphones, referrals from membership lists, social networks, mobile games, banner advertisements, mail-based recruitment campaigns and others (28,29). The survey was estimated to take 5-10 minutes to complete and all participants were compensated.”

A lot of people might not be able to use the internet or fill out an online survey because they are not tech-savvy enough. How will they be a part of the study? How can you say that the survey is representative of the whole country?

Thank you for raising this point. This survey was only accessible online. As mentioned above, the survey is representative of the whole country based only the criteria of age, race, and ethnicity. 

Notably, according to Pew Research Center in 2021, 75% of US adults over 65 years of age have internet access. (1)

1. Perrin A, Atske S. 7% of Americans don’t use the internet. Who are they? [Internet]. Pew Research Center. [cited 2023 Jun 13]. Available from: https://www.pewresearch.org/short-reads/2021/04/02/7-of-americans-dont-use-the-internet-who-are-they/

Inclusion and exclusion criteria should be more specific

The only inclusion/exclusion criteria are below, as provided in the Materials and Methods section:

“Participants were required to be at least 65 years of age, located in the United States, and fluent in the English language.”

This survey is focusing on the efficiency of 5 minutes and 45 minutes of strength training for US older adults; why not 10/15/20 and 45 minutes?

Thank you for highlighting this. There are infinitely many possible lengths of strength training sessions and frequencies that are possible to test. Given the rise in popularity of brief workouts, a 5-minute option was selected to draw a clear distinction between what is typically offered for older adults (i.e., 30-45 mins). Notably, this is mentioned as a limitation, as illustrated in the two quotes below from the discussion section:

“This study is not intended to suggest that a 5-minute daily strength training workout is the best strength training format for older adults or that it is better than a 45-minute program completed three times per week. Further exploration into alternative/additional program details, the number of options presented, and how those programs are described is required.”

“This study has several limitations. First, only two program options were evaluated; information about any alternative program options that may be preferable or elicit higher self-efficacy evaluations is not available.”

Discuss the generalizability of the study findings

This study is generalizable insofar as a cross-sectional survey has the capability to be. It is an evaluation of preferences and self-efficacy for two strength training programs in a sample that is nationally-representative of the sex, race, and ethnicity of the US population over the age of 65. Important next steps have been suggested in the Discussion section, as quoted below:

“This is a preliminary investigation into asking for program evaluations using a known predictor of the target behavior (in this case, barrier self-efficacy for physical activity) instead of preference. This study is not intended to suggest that a 5-minute daily strength training workout is the best strength training format for older adults or that it is better than a 45-minute program completed three times per week. Further exploration into alternative/additional program details, the number of options presented, and how those programs are described is required. Most importantly, investigation also needs to be done to evaluate differences in actual behavioral uptake, maintenance, and outcomes that arise from two comparative interventions: one designed using participant preferences and one designed using a known predictor of maintenance.”

This study should acknowledge the potential bias of the study.

Potential biases are discussed in a limitations paragraph, found in the Discussion section, as below:

“This study has several limitations. First, only two program options were evaluated; information about any alternative program options that may be preferable or elicit higher self-efficacy evaluations is not available. Second, the relevance of the difference in self-efficacy, though large (Cohen’s d = 1.3), is unclear. While one program was often rated higher than the other and dichotomized as the program for which self-efficacy was higher, the magnitude of that difference varied by participant. Finally, the impact of preference and self-efficacy evaluations was not evaluated by assessing actual behavior uptake, maintenance, or outcomes. It remains unclear if there is a behavioral or outcome-based difference between preference and self-efficacy-designed programs.”

Reviewer #1 Comments

Thank you for inviting me to review this paper. This paper is well written and very interesting and important for the readers. However, there are some minor changes required to improve quality of this paper.

Introduction

The introduction section should be elaborated by reflecting the significance and novelty of the study.

Thank you for this suggestion. The following section of the Introduction has been edited to address this feedback:

“The goal of most physical activity programming is long-term engagement in the behavior, or maintenance, as repeated performance is required to reap the benefits associated with physical activity. Identifying factors associated with maintenance of regular physical activity – and in fact even defining the phenomenon of “maintenance” itself – remains an active pursuit of both research and clinical practice (21). However, of what is known, self-efficacy (SE; the level of confidence in one’s capability to engage in the behavior) is to-date perhaps the single most well-established predictor of physical activity engagement and maintenance according to systematic reviews and meta-analyses (22–25). As such, designing programs that maximize known predictors of long-term physical activity engagement such as self-efficacy may promote long-term behavior performance. One novel method by which this might be accomplished is to use the evaluation of program characteristics rated using a known predictor of physical activity maintenance (i.e., self-efficacy) in place of the traditional a rating of preference (26).”

Methodology

What were the exclusion criteria? Who did the survey?

The statistical analysis section should reflect all the tests used.

Thank you for requesting this clarification. The criteria for inclusion are state in the Materials and Methods section, as below:

“Participants were required to be at least 65 years of age, located in the United States, and fluent in the English language.”

Participant description is available in the Results section, as below, and also in Table 1:

“A demographic summary of the 611 participants that completed the survey is provided in Table 1. Of note, the mean participant age was 72 years. Approximately half of participants were female. Racial and ethnic distributions were representative of the US population over the age of 65 (32).”

The statistical tests used are identified in the Materials and Methods section, as below:

“Participants were grouped by their expressed strength training program preference. Additionally, two mean self-efficacy scores, one for the 5-minute daily program and one for the 45-minute program performed three days per week, were then calculated for each participant. These two scores were then compared; where applicable, the program for which participants reported higher SE was identified as the program for which the participant had the most self-efficacy. The distribution of these two variables (program preference and higher/lower/equal self-efficacy) were then cross-tabulated for comparison.”

Discussion

In discussion section, focus more on potential implications of your study's findings and add future research direction based on current findings.

Potential implications have been mentioned in the Discussion section, as below:

“…soliciting evaluations of possible program options on known predictors of the target behavior (in this case self-efficacy for physical activity maintenance) instead of preference may lead to more effective decision-making by intervention designers targeting physical activity maintenance.

Future directions are suggested in the Discussion section as below:

“Further exploration into alternative/additional program details, the number of options presented, and how those programs are described is required. Most importantly, investigation also needs to be done to evaluate differences in actual behavioral uptake, maintenance, and outcomes that arise from two comparative interventions: one designed using participant preferences and one designed using a known predictor of maintenance.”

Conclusion

Clearly state the contributions of your study to the field and highlight any novel aspects of your research. Future work should be mentioned in discussion section. 

Thank you for this suggestion. Future directions have been removed from the Conclusion section. Implications have been added as below:

“In this novel study directly comparing preferences and self-efficacy, adults over the age of 65 years reported preferring a 5-minute daily strength training program compared to a 45-minute program performed three times per week. Most participants also reported higher self-efficacy for that program. However, 1 in 5 older adults preferred a program for which they had lower self-efficacy than the alternative; nearly all of these participants preferred the 45-minute program but had lower self-efficacy for it. This evidence suggests that expressed preferences for physical activity programming may not always reflect the program most likely to be maintained. Soliciting evaluations of possible program options on known predictors of the target behavior (in this case self-efficacy for physical activity maintenance) instead of preference may lead to more effective decision-making by intervention designers targeting physical activity maintenance.”

Language and Writing Style

The manuscript requires careful proofreading to eliminate grammatical errors and improve sentence structure. Clarity and precision in language will s

---

## [Decision Letter · Decision Letter 1]

16 Apr 2024

Comparing preferences to evaluations of barrier self-efficacy for two strength training programs in US older adults

PONE-D-24-04115R1

Dear Dr. Jordan D Kurth

We’re pleased to inform you that your manuscript has been judged scientifically suitable for publication and will be formally accepted for publication once it meets all outstanding technical requirements.

Kind regards,

Sohel Ahmed, BPT, MPT, MDMR

Academic Editor

PLOS ONE

Reviewers' comments:

Reviewer's Responses to Questions

**Comments to the Author**

1. If the authors have adequately addressed your comments raised in a previous round of review and you feel that this manuscript is now acceptable for publication, you may indicate that here to bypass the “Comments to the Author” section, enter your conflict of interest statement in the “Confidential to Editor” section, and submit your "Accept" recommendation.

Reviewer #1: All comments have been addressed

Reviewer #2: All comments have been addressed

2. Is the manuscript technically sound, and do the data support the conclusions?

Reviewer #1: Yes

Reviewer #2: Yes

3. Has the statistical analysis been performed appropriately and rigorously? 

Reviewer #1: Yes

Reviewer #2: Yes

4. Have the authors made all data underlying the findings in their manuscript fully available?

Reviewer #1: Yes

Reviewer #2: No

5. Is the manuscript presented in an intelligible fashion and written in standard English?

Reviewer #1: Yes

Reviewer #2: Yes

6. Review Comments to the Author

Reviewer #1: (No Response)

Reviewer #2: (No Response)

7. PLOS authors have the option to publish the peer review history of their article (what does this mean?). If published, this will include your full peer review and any attached files.

Reviewer #1: No

Reviewer #2: **Yes: **Sahar Boozari

---

## [Editor Report · Acceptance letter]

26 Apr 2024

PONE-D-24-04115R1 

PLOS ONE

Dear Dr. Kurth, 

I'm pleased to inform you that your manuscript has been deemed suitable for publication in PLOS ONE. Congratulations! Your manuscript is now being handed over to our production team.

Kind regards, 

on behalf of

Dr. Sohel Ahmed 

Academic Editor

PLOS ONE